# Possible Use of Minocycline in Adjunction to Intranasal Esketamine for the Management of Difficult to Treat Depression following Extensive Pharmacogenomic Testing: Two Case Reports

**DOI:** 10.3390/jpm12091524

**Published:** 2022-09-16

**Authors:** Matteo Marcatili, Riccardo Borgonovo, Noemi Cimminiello, Ranieri Domenico Cornaggia, Giulia Casati, Cristian Pellicioli, Laura Maggioni, Federico Motta, Chiara Redaelli, Luisa Ledda, Federico Emanuele Pozzi, Michaela Krivosova, Jessica Pagano, Roberto Nava, Fabrizia Colmegna, Antonios Dakanalis, Alice Caldiroli, Enrico Capuzzi, Beatrice Benatti, Bernardo Dell’Osso, Francesca Bertola, Nicoletta Villa, Alberto Piperno, Silvia Ippolito, Ildebrando Appollonio, Carlo Sala, Luciano Conti, Massimo Clerici

**Affiliations:** 1Psychiatric Department, San Gerardo Hospital, ASST Monza, 20900 Monza, Italy; 2Department of Medicine and Surgery, University of Milano Bicocca, 20900 Monza, Italy; 3Department of Neurology, San Gerardo Hospital, ASST Monza, 20900 Monza, Italy; 4Biomedical Centre Martin, Jessenius Faculty of Medicine in Martin, Comenius University in Bratislava, 03601 Martin, Slovakia; 5National Research Council Neuroscience Institute, 20100 Milan, Italy; 6Psychiatry Unit, Department of Biomedical and Clinical Sciences “Luigi Sacco”, University of Milan, 20100 Milan, Italy; 7CRC “Aldo Ravelli” for Neurotechnology and Experimental Brain Therapeutics, University of Milan, 20100 Milan, Italy; 8Cytogenetics and Medical Genetics Unit, Centre for Disorders of Iron Metabolism, San Gerardo Hospital, ASST Monza, 20900 Monza, Italy; 9Clinical Chemistry Laboratory, San Gerardo Hospital, ASST Monza, 20900 Monza, Italy; 10Laboratory of Stem Cell Biology, Department of Cellular, Computational and Integrative Biology (CIBIO), University of Trento, 38122 Trento, Italy

**Keywords:** treatment resistant depression (TRD), difficult to treat depression (DTD), minocycline, esketamine, ketamine, pharmacogenetics, major depressive disorder (MDD)

## Abstract

The advent of intra-nasal esketamine (ESK), one of the first so called *fast-acting antidepressant*, promises to revolutionize the management of treatment resistant depression (TRD). This NMDA receptor antagonist has proven to be rapidly effective in the short- and medium-term course of the illness, revealing its potential in targeting *response* in TRD. Although many TRD ESK responders are able to achieve remission, a considerable portion of them undergo a metamorphosis of their depression into different clinical presentations, characterized by instable responses and high recurrence rates that can be considered closer to the concept of *Difficult to Treat Depression* (DTD) than to TRD. The management of these DTD patients usually requires a further complex multidisciplinary approach and can benefit from the valuable contribution of new personalized medicine tools such as therapeutic drug monitoring and pharmacogenetics. Despite this, these patients usually come with long and complex previous treatments history and, often, advanced and sophisticated ongoing pharmacological schemes that can make the finding of new alternative options to face the current recurrences extremely challenging. In this paper, we describe two DTD patients—already receiving intranasal ESK but showing an instable course—who were clinically stabilized by the association with minocycline, a semisynthetic second-generation tetracycline with known and promising antidepressant properties.

## 1. Introduction

The advent of intra-nasal esketamine (ESK), one of the first antidepressants, together with brexanolone, a so-called *fast-acting antidepressant*, promises to revolutionize the management of treatment resistant depression (TRD). This N-methyl-D-aspartate (NMDA) receptor antagonist has proven to be rapidly effective in the short- and medium-term course of the illness [1,2,3,4,5], revealing its potential in targeting *response* in TRD, i.e., in patients who have not responded to at least two different treatments with antidepressants in the context of the same major depressive episode. Nevertheless, the level of proof of ESK efficacy in the long-term course of major depressive disorder (MDD) remains low [6], and further research is needed. Although many TRD ESK-responder patients might achieve remission, a considerable portion of them could assist in the metamorphosis of their depression into different clinical presentations, characterized by instable responses and high recurrence rates that can be considered closer to the concept of *Difficult to Treat Depression* (DTD) than to TRD. Recently described as “depression that continues to cause significant burden despite usual treatment effort” [7], DTD is characterized by impaired daily function, poor quality of life, and a wide genetic and etiological heterogeneity [8]. As we will show later, the management of these DTD patients can require a further complex and multidisciplinary approach and can benefit from the valuable contribution of some personalized medicine tools such as therapeutic drug monitoring (TDM) and pharmacogenetics. Despite this, these patients usually come with long and complex previous treatments history and, often, advanced and sophisticated ongoing pharmacological schemes that can make the finding of new alternative options to face the current recurrences extremely challenging. So, what can be done with these DTD patients already being treated with ESK (plus SSRI/SNRI) who exhibit instable response patterns characterized by high recurrence rates?

We will focus on minocycline (MIN), a semisynthetic second-generation tetracycline, known for its antidepressant properties since 1996 [9] and the object of study of different clinical trials [10,11,12,13,14] and meta-analysis [15]. Its antidepressant mechanisms of action have not been fully elucidated but, thanks to its good penetration through the blood–brain barrier, MIN is thought to exert a local neuroprotective effect through a complex mixture of anti-inflammatory, anti-apoptotic, anti-oxidative, glutamatergic, and monoaminergic activities [16].

In this paper, we will describe the cases of two DTD patients—already receiving intranasal ESK but showing an instable course of the illness—clinically stabilized by the association with MIN.

## 2. Methods

The following two outpatients from our Treatment-Resistant Disorder Clinic were selected for this case report according to their diagnosis of TRD. Both patients were experiencing a further relapse while being on intranasal esketamine.

In line with the protocol of our clinic, the patients underwent an extensive assessment, including anamnesis and a detailed medication history collection (specifically investigating dosages, intervals, and reasons for the changes/interruptions). The psychiatric interviews also included a psychometric assessment using Montgomery-Asberg Depression Rating (MADRS), Hamilton Anxiety (HAM-A), Hamilton Depression (HAM-D), Brief Psychiatric Rating (BPRS), and Clinical Global Impression (CGI) scales (see the legends for details). Moreover, innovative clinical tools such as therapeutic drug monitoring and pharmacogenetic testing were applied.

Further treatment strategies, for example using an off-label medication such as minocycline, were made according to the literature review and were subsequently discussed in detail and approved by the patients. They provided a written informed consent for both the off-label treatment and case publication. Particular emphasis was placed on the possible side effects and on the uncertain acute as well as long-term benefits because of limited scientific evidence.

## 3. Case Presentation 1

We present the case of a 73-year-old married woman, T.R., diagnosed with MDD. She had a positive psychiatric family history, because her mother and younger sister suffered from MDD. The onset of her depressive symptomatology was observed in 2016, at the age of 68. At the onset, the patient experienced the classic symptoms of depression (thymic deflection, abulia, anhedonia, apathy) associated with cognitive impairment and somatic neurovegetative symptoms.

Initially, the symptoms were managed by the primary care physician with paroxetine 40 mg/day for 4 months. Due to its ineffectiveness, paroxetine was switched to duloxetine, maintained for the following 4 months, and increased to a dosage of up to 90 mg/day.

In 2017, because T.R. did not show any response to duloxetine, she was twice hospitalized in a psychiatric facility for two different long-stays for a total duration of 3 months; she was treated with vortioxetine up to a dosage of 20 mg/day during the stay without showing any improvement. In the context of the second long-stay, due to the persistent presence of cognitive symptoms (e.g., severe memory deficit, executive dysfunction, spatiotemporal disorientation, and difficulties performing calculations and a Mini–Mental State Examination (MMSE) = 15, suggestive of moderate global cognitive impairment), a primary cognitive impairment was hypothesized, and she was thus referred to an adult day-center for people suffering from neurodegenerative disorders. However, brain Magnetic Resonance Imaging (MRI) was substantially normal and a positron emission tomography (PET) with fluorodeoxyglucose (FDG) (2019) showed only a very slight reduction in parietal and frontal cortex metabolism.

Consequently, in early 2018, T.R. had a new psychiatric consultation where the diagnostic hypothesis was changed to depression-related cognitive dysfunction (Pseudodementia), and a new treatment with the tricyclic antidepressant clomipramine was started and titrated up to 150 mg/day. Finally, she showed a response and, after a few months, a complete remission that lasted until September 2020 when, despite the clomipramine titration up to 225 mg/day and the introduction of two adjunctive treatments (amisulpride 50 mg/day, SAMe 800 mg/day) she experienced a dramatic relapse and was diagnosed with TRD. For this reason, she was referred to our Treatment-Resistant Disorder Clinic. The multiple pharmacological trials in her clinical history are summarized in Table 1. Mood variations with treatment adjustments are shown in Figure 1.

During the first diagnostic interview at our Clinic, T.R. presented with severe melancholic depression with significant psychomotor retardation. In detail, depressed mood with persistent anhedonia, lack of will, and severe somatic anxiety levels (e.g., tremor, dizziness, and tachycardia) were present. The emotional lability was characterized by frequent and unexpected crying fits. Suicidal ideation was present. The cognitive impairment was remarkable: the patient reported the incapability of doing ordinary tasks (e.g., personal hygiene, cooking, reading, and watching TV) due to a significant loss of memory and concentration. Moreover, she had issues in recalling names and she was only partially oriented in time and space. The motor impairment was visible during her walk, and the postural balance was uncertain.

When T.R. came to our clinic, the therapy consisted of clomipramine 225 mg/day, SAMe 800 mg/day, amisulpride 50 mg/day.

In line with our protocol, psychiatric interview, psychometric assessment (Table 2), TDM (Table 3) and pharmacogenetic analysis were performed (Table 4).

Considering the psychiatric rating scales scores (Table 2) and the clinical interviews, the patient was diagnosed as affected by a severe major depressive episode in the context of a confirmed diagnosis of TRD. According to European Medicines Agency (EMA) criteria, the patient was suitable for an intranasal esketamine treatment. Sertraline 50 mg/day was added as a combination strategy and pregabalin was titrated up to 225 mg/day to control anxiety. In addition, considering the high blood clomipramine level (Table 3), the dosage was reduced to 150 mg/day.

As shown in Figure 2, since the beginning of the induction schedule with esketamine, T.R. showed a rapid improvement, followed by recurrent relapses. The single relapses were coped by changes in ongoing therapy and by a deviation from the standard pattern of esketamine administration.

In fact, during the first relapse, at the eighth administration, it was decided to increase the clomipramine dose back to 225 mg/day. Subsequently, during the second relapse, we introduced lithium sulfate up to 124.5 mg/day, but it was then reduced and stopped due to the patient’s intolerance. At the 29th administration, there was a new mood deflection, which was treated through a pharmacological switch from sertraline to venlafaxine up to a dose of 225 mg/day. At the fourth relapse, L-methylfolate was introduced as an augmentation strategy, in consideration of the pharmacogenetic results (it was estimated a 70% methylene tetrahydrofolate reductase, MTHFR, reduction). Later on, aripiprazole (5 mg/day) was introduced to successfully treat a new recurrence. Unfortunately, approximately 5 months later, yet another severe depressive episode occurred. Given the current complex antidepressant scheme (combination of the glutamatergic esketamine plus the SNRI venlafaxine plus the tryciclic clomipramine, all at high dosages) strongly impacting on all monoaminergic and glutamatergic systems, we hypothesized that a further boost on the same systems would not have been of any further benefit. We therefore considered an accessible antidepressant compound with a possible complementary and synergic mechanism of action, and minocycline was chosen as a convincing option for the treatment of this current relapse.

MIN was introduced at a dosage of 200 mg/day (100 mg BID). One week after minocycline introduction, the depressive symptoms (sadness, apathy, and lack of concentration) partially reverted (MADRS = 16). Two weeks after the beginning of the treatment, there was a clear clinical improvement with a drastic reduction in all psychiatric rating scores (MADRS = 6), and no side effects were noticed. No signs of dysbiosis were observed, while blood levels of erythrocyte sedimentation rate (ESR) and C-reactive protein (CRP) have remained within the reference range.

The clinical improvement had been gradual but consistent; a complete remission was achieved after 7 weeks of treatment (MADRS = 0), together with cognitive improvement (MMSE = 28) with persistent mild executive dysfunction, as shown by Frontal Assessment Battery = 8/18.

## 4. Case Presentation 2

The second case concerns L.B., a 61-year-old man, diagnosed with MDD. The patient has a family psychiatric history suggestive of mood disorders because his mother suffered from bipolar disorder and his daughter suffers from MDD and anorexia nervosa.

Regarding the psychopathological onset, L.B. reported his first major depressive episode in 1999, at the age of 38 when he started experiencing typical depressive symptoms such as apathy, abulia, social withdrawal, and inability to work. This episode required a long hospitalization in a private clinic where he was treated with a complex approach (imipramine 50 mg/day, lithium carbonate 600 mg/day, and carbamazepine 400 mg/day combined with sleep deprivation and light therapy). Subsequently, the patient remained in a state of clinical well-being until 2015 when, at the age of 53, he was diagnosed with a severe dilated cardiomyopathy and implanted with a cardioverter defibrillator (ICD) and a pacemaker (PM). Due to its possible cardiotoxicity, lithium was stopped. Since then, L.B. experienced multiple severe relapses that required psychiatric hospitalizations, most of them for suicide attempts. Unfortunately, from 2017 to 2021 many appropriate treatments were attempted without obtaining any substantial and stable remission. The multiple pharmacological trials in his clinical history are summarized in Table 5. Mood variations with treatment adjustments are shown in Figure 3.

In March 2021, due to the frequent relapses and the documented treatment-resistance, the patient was referred to our Treatment-Resistant Disorder Clinic, where he was diagnosed with Difficult-to-Treat Depression (DTD). When L.B. came to our clinic, his therapy consisted of sertraline 100 mg/day, quetiapine XR 300 mg/day, and quetiapine 25 mg/day. In line with our protocol, psychiatric interview, psychometric assessment (Table 6) and pharmacogenetic analysis were performed (Table 7).

During the first diagnostic interview, his mood appeared severely depressed with multiple cognitive symptoms (e.g., short-term memory impairment and poor concentration), psychomotor retardation, emotional lability, frequent crying fits and, in addition, an excessive sense of guilt and hopelessness regarding his psychic condition (MADRS = 36). Despite the heart disease, based on its past effectiveness, lithium was re-introduced in order to obtain a clinical stabilization. Over the course of treatment, a mood stabilization was observed (MADRS = 10) and a remarkable improvement in daily psychosocial functioning occurred.

Nonetheless, residual depressive symptoms, mostly the cognitive ones (e.g., lack of concentration and poor planning task abilities), persisted. Thus, the therapy was modified by adding amisulpride (50 mg/day), SAMe (400 mg/day), and choline alfoscerate (1200 mg/day). These therapeutic measures produced some further improvement in the residual symptoms (MADRS = 7).

In May 2021, putatively reactive to important external factors (mournful events), a depressive relapse occurred (MADRS = 30). Amisulpride was suspended while nortriptyline was introduced (titrated up to 100 mg/day) and, at the same time, sertraline was titrated up 150 mg/day.

In July 2021, after a 6-week period of treatment with the tricyclic antidepressant and no response observed, the patient started the administration of intranasal ESK. Ensured a good tolerance to 56 mg, after 5 days the dose was increased to 84 mg twice a week for the entire induction phase. At the end of the first ESK treatment month an impressive reduction of depressive symptoms was observed (MADRS = 6). The weekly administration was continued at 84 mg/week through the maintenance phase.

However, at the beginning of the maintenance phase, an important mood deflection was observed (MADRS = 35). Bupropion was subsequently titrated up to 300 mg/day as a further augmentation strategy, sertraline was reduced to 50 mg/day, and quetiapine was gradually stopped.

Despite the polypharmacotherapy, no clinical benefit was observed, and the patient was still anergic and hypobulic, with severe cognitive dysfunction (lack of attention and concentration). Specifically in consideration of the importance of these cognitive dimensions in the patient syndrome, at the end of August atomoxetine 40 mg/day was introduced and bupropion was gradually stopped. Furthermore, on the basis of the estimated MTHFR activity reduction by approximately 35% inferred by the pharmacogenetic analysis, L-methylfolate (15 mg/day) was added to the therapy. Two weeks later, a rapid improvement was observed (MADRS = 8) and in four weeks a complete regression of the depressive symptomatology was achieved (MADRS = 0). This oral treatment modulation ensured a 4-month period of well-being until December 2021, when the last major depressive episode occurred. At that time the symptoms were particularly severe (MADRS = 46): mood deflection, apathy, abulia, anhedonia, somatic anxiety, sense of guilt, suicidal thoughts, and hyporexia associated with weight loss (about 3–4 kg). Atomoxetine dosage was increased to 80 mg/day, but the patient’s condition did not improve. We therefore took some time to discuss with the patient and his family all the treatment options, because many pharmacological attempts had been made and many neuromodulation techniques, such as repetitive transcranial magnetic stimulation (rTMS) or electroconvulsive therapy (ECT), were contraindicated due to the presence of the ICD with PM.

Among the different pharmacological and non-pharmacological treatment options proposed, the shared choice fell on minocycline as a result of clinical reasoning similar to what has been described for case 1. MIN was therefore introduced at the dosage of 200 mg/day, divided into two administrations of 100 mg per day, while the ESK maintenance scheme was continued.

One week after the introduction of this tetracycline, there was an initial improvement in depressive symptoms (MADRS = 32), accompanied by the presence of subjective clinical benefit. Neither side effects nor dysbiosis phenomena were reported. Blood levels of erythrocyte sedimentation rate (ESR) and C-reactive protein (CRP) remained within the reference range.

Two weeks after the beginning of the treatment, a further improvement was observed (MADRS = 30), especially in the areas of abulia and anergy.

At the third and fourth week after the introduction of minocycline, a huge improvement of the clinical condition was observed, with an almost complete regression of depressive symptoms (MADRS = 6). The patient reported a significant positive change in quality of life with mood stability, adequate volitional, and planning drives.

The observed improvement was maintained at subsequent clinical examinations. Mood variations throughout the ESK treatment with therapy adjustments are shown in Figure 4. The patient can be currently (in the eighth week after minocycline treatment) considered in remission, in a state of self-reported “well-being”.

## 5. Discussion

We presented two complex cases of DTD where ESK appeared to be necessary to obtain response, otherwise very difficult to achieve, but non-sufficient to warrant a stable remission.

In the management of such difficult cases, personalized medicine tools, such as TDM and pharmacogenetic analyses, can bring great benefit, due to their capacity to provide additional valuable information that can be useful both to conceptualize the case (i.e., biological contributors to treatment resistance) and to optimize the treatment.

Considering the results of pharmacogenetic analysis, relatively to pharmacokinetic genes, T.R. has a status of CYP2B6 poor, CYP2D6 intermediate, and CYP2C19 rapid metabolizer. Clomipramine (CLO), which has been used during the entire antidepressant intervention of the patient and improved the condition, is primarily metabolized by CYP2C19 to the active metabolite desmethylclomipramine (DCLO), which is subsequently metabolized by CYP2D6 [24]. In previous studies, no clear correlation was found between CYP2C19 genotype and clomipramine metabolism [24], but intermediate function of CYP2D6 resulted in increased serum levels of the active moiety, i.e., serum concentration of CLO + DCLO [25]. This is in accordance with our findings of the drug level above the therapeutic range at patient’s first visit. Regarding other antidepressants, rapid metabolism of sertraline by CYP2C19 could cause the failure of such treatment, and intermediate metabolism of paroxetine by CYP2D6 could increase the serum levels as well as the higher probability of side effects in this patient [26]. At the therapeutic concentrations, the major CYP450 isoform involved in N-demethylation of ketamine to norketamine in vitro is CYP2B6 [27]. However, there was not found any significant difference between CYP2B6 genotypes in ketamine metabolism in vivo [28], so being a CYP2B6 poor metabolizer, such as T.R, does not seem to substantially impact ESK bioavailability or contraindicate ESK intranasal treatment.

By contrast, reduced enzyme activity of CYP2C9 was found in patient L.B. This CYP450 isoform is partially involved in the metabolism of different SSRIs and, to a lesser extent, to ketamine N-demethylation [29]. Regarding fluoxetine metabolism, for instance, a study found a potential influence of CYP2C9 genotypes on plasma concentration of active moiety (fluoxetine plus norfluoxetine), but a significant difference was not found between the individuals with one and both mutated alleles in this gene [30]. To conclude, the pharmacokinetic gene variations profile of the patient L.B. does not seem to fully explain his unresponsiveness to various treatment strategies.

Generally, the outcomes of pharmacodynamic gene studies in psychiatric disorders are of low evidence, and the relative literature can be considered still pioneeristic, with limited and sparse data that cannot be easily translated to the clinic. Nevertheless, the pharmacodynamic gene variations emersed in these two cases could open up some interesting insights. Both patients present a single nucleotide polymorphism (SNP) in the gene *GABRA6* (rs3219151) that had been associated with increased vulnerability to MDD and anxiety disorders [19,31]. Another SNP in gene *GRIK4* (rs1954787)*,* Glutamate Ionotropic Receptor Kainate Type Subunit 4, has been previously associated with a worse response to first choice antidepressants SSRIs and SNRIs [20] and could partially contribute to non-responsiveness to paroxetine and duloxetine in the T.R. case and to citalopram, duloxetine, fluoxetine, venlafaxine, and sertraline in the L.B. case.

*COMT* polymorphism (rs4680) has already been widely debated, and meta-analyses were published in relation to a potential susceptibility for MDD [32] and ECT treatment response [21]. The link was also studied between rs4680 and response to alternative strategies such as sleep deprivation combined with light therapy in bipolar depressed patients [33]. The last study observed a better response in patients carrying the A allele, which is consistent with our case, as L.B. improved and remained remitted for a long period after such treatment combined with antidepressants. ECT therapy was not performed in these two cases. A higher probability of antipsychotic- and antidepressant-related side effects has lately been associated with genetic variants in *HTR2C* and *HTR2A* genes [22,23,34,35,36]. In our patients, we have not observed SSRI-related side effects nor weight gain related to antipsychotics. Finally, in both our patients a putative reduction in MTHFR activity, an essential enzyme in monoamine biosynthesis [37], was detected. Previous studies showed improved antidepressant response with folate supplementation in patients with such polymorphisms [17,18]. When L-methylfolate augmentation was introduced in both our patients, an improvement of their mood stabilization in that period was recorded.

Moreover, we mentioned a large number of treatment-specific strategies that could be employed in such cases and are extensively described elsewhere [38,39], such as antidepressant dose escalation, switching to another antidepressant, combination of two or more antidepressant and, finally, augmentation of the ongoing antidepressant trial with compounds of other substance classes. Many of these strategies were considered for our patients and, when implemented, they sometimes appeared to be able to temporarily improve the symptoms but, unfortunately, not to achieve a stable remission. For these reasons, when both the last relapses invariably occurred, we decided to follow a different approach and MIN was chosen as a further augmentation strategy. Its antidepressant mechanism of action is not completely clear, but it probably relies on its ability to inhibit kynurenine and p-38 pathways. In facts, in the context of the inflammatory hypothesis of depression, inflammation could elicit, through the kynurenine pathway, the activation of 2,3-dioxygenase (IDO), an enzyme implicated in the metabolism of the serotonin precursor tryptophan, leading to a reduction of serotonin levels [40]. Furthermore, inflammation processes could engage the p38 pathway and increase the expression of the serotonin transporter, leading to a reduction of this neurotransmitter in the synaptic cleft [41,42,43]. Furthermore, some studies have shown that minocycline is able to modulate glutamatergic pathways because minocycline reduces glutamate neurotransmission [44], probably by inhibiting microglial activation and by enhancing glial glutamate transport [45,46]. Moreover, MIN was found to modulate the GluR1 subtype of the α-amino-3-hydroxy-5-methyl-4-isoxazolepropionic acid (AMPA) receptor, both in vitro and in vivo [47]. Increased GluR1 phosphorylation (and AMPA receptor potentiation) was associated with antidepressant effects [47].

In addition, the combination of subthreshold doses of minocycline with glutamate receptor antagonists such as dizocilpine (NMDA receptor antagonist) have previously produced antidepressant-like effects in animal models [48]. The synergic effect might be explained by the ability of MIN to reduce NMDA- and glutamate-induced neurotoxicity, thus preventing neuronal death [49].

### Limitations

The present manuscript describes the use of esketamine and minocycline combination therapy in only two DTD patients and, thus, no clear conclusions can be drawn. Further studies on a larger population should be made in order to rigorously verify the clinical benefit of this augmentation strategy in such patients. Another limitation is the relatively short follow-up period of only 7–8 weeks. Longer studies to assess the remission maintenance are warranted.

## 6. Conclusions

The management of DTD patients requires a complex multidisciplinary approach and can benefit from the valuable contribution of new personalized medicine tools such as therapeutic drug monitoring and pharmacogenetics. Our patients, already receiving intranasal ESK but showing an instable course, were clinically stabilized by additional off-label treatment with MIN, a tetracycline with known and promising antidepressant properties. Beyond all the interesting speculation on MIN mechanisms of actions and its possible synergistic activities to glutamatergic compounds derived from animal studies, further research on human models is needed to shed light on these issues. For this purpose, studies employing human induced pluripotent stem cell (hiPSC) technology [50] in modelling TRD could represent a valid option for overcoming the limitations derived from animal models and provide innovative disease-relevant patient-specific in vitro models that can parallel promising clinical outcomes such as those described in these case reports.

For this reason, these patients have been included in an ongoing research project aimed at modeling TRD and treatment response or resistance at the molecular level through the cellular reprogramming granted by hiPSC technology. This research approach may hopefully contribute to a further clarification of MIN antidepressant mechanisms of action.

With regard to the clinical effectiveness, bigger and longer studies are warranted to further confirm the benefit of MIN and ESK in the treatment of DTD patients.

## Figures and Tables

**Figure 1 jpm-12-01524-f001:**
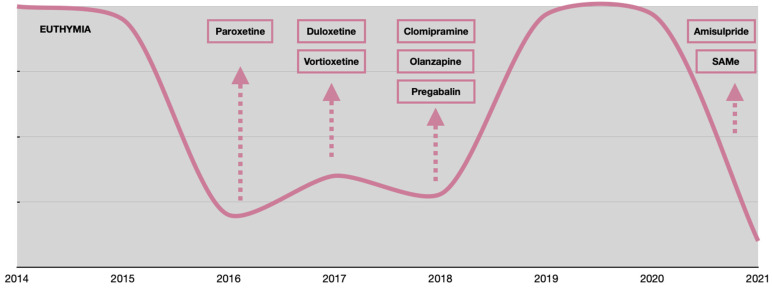
Mood variations from 2014 to 2021.

**Figure 2 jpm-12-01524-f002:**
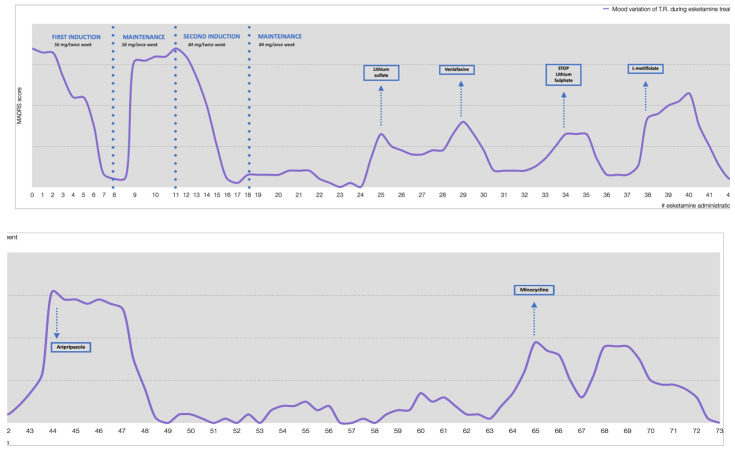
Mood variation and pharmacologic modulation during esketamine treatment.

**Figure 3 jpm-12-01524-f003:**
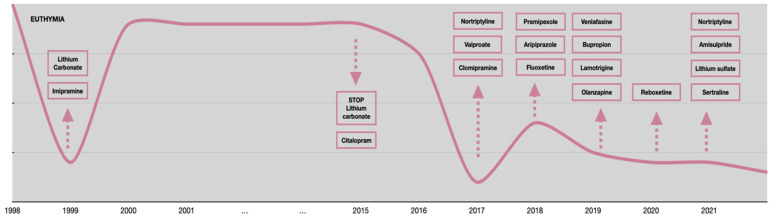
Mood variations from 1998 to 2021.

**Figure 4 jpm-12-01524-f004:**
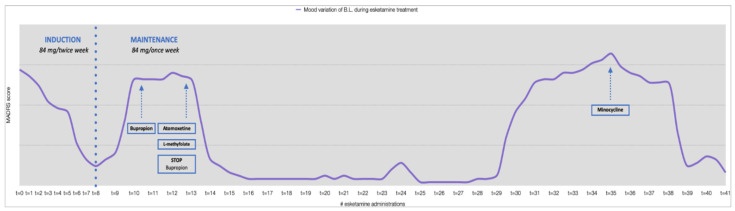
Mood variation and pharmacologic modulation during ESK treatment.

**Table 1 jpm-12-01524-t001:** Treatment history.

YEAR	MEDICATION	DAILY DOSE	TREATMENT DURATION	COMMENT
2016	Paroxetine	40 mg	4 months	No Response
2017	Duloxetine	90 mg	4 months	No Response
2017	Vortioxetine	20 mg	1 Year	No Response
2018	Olanzapine	2.5 mg	3 months	No Response
2018	Clomipramine	150 mg	Ongoing	Response
2018	Pregabalin	300 mg	Ongoing	Response
2020	Amisulpride	50 mg	2 weeks	No Response
2020	SAMe	800 mg	2 months	No Response

**Table 2 jpm-12-01524-t002:** Psychometric assessment at first consultation.

RATING SCALE	SCORE	RANGE
**BPRS**	34	18–126
**MADRS**	33	0–54
**HAM-D**	20	0–68
**CGI-S**	5	1–7

BPRS—Brief Psychiatric Rating Scale (31–40—mildly ill, 41–52—moderately ill, 53–126—markedly ill). MADRS—Montgomery-Asberg Depression Rating Scale (0–6—normal/symptom absent, 7–19—mild depression, 20–34—moderate depression, 34–54—severe depression). HAM-D—Hamilton Depression Rating Scale (0–7—no depression, 8–16—mild depression, 17–23—moderate depression, 24–68—severe depression). CGI-S—Clinical Global Impression-Severity Scale (1—normal, not at all ill, 2—borderline ill, 3—mildly ill, 4—moderately ill, 5—markedly ill, 6—severely ill, 7—among the most extremely ill).

**Table 3 jpm-12-01524-t003:** Therapeutic Drug Monitoring of clomipramine at first consultation.

Medication	Blood Level	Therapeutic Range
**Clomipramine**	>567.0 ng/mL	230.0–450.0 ng/mL

**Table 4 jpm-12-01524-t004:** Pharmacogenetic analysis.

PHARMACODYNAMIC GENES	GENOTYPE	COMMENTS AND REFERENCES
**MTHFR** rs 1801133 rs 1801131	T/T (wild-type) A/A (homozygote)	**Reduced activity by 70%.** *Jha et al., 2016 [17];Mech & Farah, 2016 [18]*
**GABRA6** rs 3219151	T/T (homozygote)	**Increased vulnerability to MDD and anxiety disorder** *Gonda et al., 2017 [19]*
**GRIK4** rs 1954787	T/T (homozygote)	**Worse response to SSRIs and SNRIs** *Kawaguchi et Glatt, 2017 [20]*
**COMT** rs 4680	A/A (homozygote)	**Worse response to ECT** *Tang et al., 2020 [21]*
**HTR2C** rs 3813929	C/T (heterozygote)	**Reduced weight gain risk** *Chen et al., 2020 [22]*
**HTR2A** rs 6313 rs 6311	G/A (heterozygote) C/T (heterozygote)	**Less probable SSRI-related side effects** *Wan et al., 2021 [23]*
**PHARMACOKINETIC GENES**	**GENOTYPE**	**METABOLIZING TYPE**
**CYP2B6**	*6/*6	**Poor metabolizer**
**CYP2C19**	*1/*17	**Rapid metabolizer**
**CYP2D6**	3 copies *4/*10	**Intermediate metabolizer**
**CYP3A5**	*1/*3	**Intermediate metabolizer**
**CYP3A4**	*1/*1	**Normal metabolizer**
**CYP2C9**	*1/*1	**Normal metabolizer**
**CYP1A2**	*1/*1F	**Inducible**
**UGT2B15*2 c253G>T**	Homozygote	**Reduced activity**
**HLA-A 3101**	Absent	**No increased risk for carbamazepine-induced hypersensitivity reactions**
**HLA-B*1502**	Absent	**No increased risk of carbamazepine-induced Stevens-Johnson syndrome**
**ABCB 1 c.3435C>T**	Heterozygote	**Reduced activity**

**Table 5 jpm-12-01524-t005:** Treatment history.

YEAR	MEDICATION	DAILY DOSE	TREATMENT DURATION	COMMENT
1999	Imipramine	100 mg	16 years intermittently	Response
1999	Lithium Carbonate	600 mg	16 years	Response
2015	Citalopram	30 mg	2 years	No Response
2017	Clomipramine	150 mg	2 months	No Response
2017	Valproate	1500 mg	6 months	No response
2017	Nortriptyline	100 mg	1 year	Response
2018	Duloxetine	120 mg	3 months	No Response
2018	Pramipexole	0,36 mg	2 months	No Response
2018	Aripiprazole	5 mg	1 month	No response
2018	Fluoxetine	30 mg	4 months	No response
2019	Venlafaxine	300 mg	3 months	No response
2019	Bupropion	300 mg	3 months	No response
2019	Lamotrigine	200 mg	1 year	No response
2019	Olanzapine	5 mg	1 year	No response
2020	Reboxetine	8 mg	1 year	No Response
2021	Quetiapine	375 mg	4 months	No response
2021	Sertraline	150 mg	Ongoing	No Response
2021	Lithium sulfate	166 mg	1 year	Response
2021	Amisulpride	100 mg	2 months	No Response
2021	Nortriptyline	50 mg	2 months	No Response

**Table 6 jpm-12-01524-t006:** Psychometric assessment at first consultation.

RATING SCALE	SCORE	RANGE
**BPRS**	40	18–126
**MADRS**	36	0–54
**HAM-D**	25	0–68
**CGI-S**	5	1–7

BPRS—Brief Psychiatric Rating Scale (31–40—mildly ill, 41–52—moderately ill, 53–126—markedly ill). MADRS—Montgomery-Asberg Depression Rating Scale (0–6—normal/symptom absent, 7–19—mild depression, 20–34—moderate depression, 34–54—severe depression). HAM-D—Hamilton Depression Rating Scale (0–7—no depression, 8–16—mild depression, 17–23—moderate depression, 24–68—severe depression). CGI-S Clinical Global Impression-Severity (1—normal, not at all ill, 2—borderline ill, 3—mildly ill, 4—moderately ill, 5—markedly ill, 6—severely ill, 7—among the most extremely ill).

**Table 7 jpm-12-01524-t007:** Pharmacogenetic analysis.

PHARMACODYNAMIC GENES	GENOTYPE	COMMENTS AND REFERENCES
**MTHFR** rs 1801133 rs 1801131	T/T (wild-type) G/A (heterozygote)	**Reduced activity by 35%.** *Jha et al., 2016 [17]; Mech & Farah, 2016 [18]*
**GABRA6** rs 3219151	C/T (heterozygote)	**Increased vulnerability to MDD and anxiety disorder** *Gonda et al., 2017 [19]*
**GRIK4** rs 1954787	T/C (heterozygote)	**Worse response to SSRIs and SNRIs** *Kawaguchi et Glatt, 2017 [20]*
**COMT** rs 4680	A/A (homozygote)	**Worse response to ECT** *Tang et al., 2020 [21]*
**HTR2C** rs 3813929	C/C (wild-type)	**Increased weight gain risk** *Chen et al., 2020 [22]*
**HTR2A** rs 6313 rs 6311	G/A (heterozygote) C/T (heterozygote)	**Less probable SSRI-related side effects** *Wan et al., 2021 [23]*
**PHARMACOKINETICS GENES**	**GENOTYPE**	**METABOLIZING TYPE**
**CYP2B6**	*1/*1	**Normal metabolizer**
**CYP2C19**	*1/*1	**Normal metabolizer**
**CYP2D6**	*2/*2*41	**Normal metabolizer**
**CYP3A5**	*3/*3	**Poor metabolizer**
**CYP3A4**	*1/*1	**Normal metabolizer**
**CYP2C9**	*2/*3	**Poor metabolizer**
**CYP1A2**	*1F/*1F	**Inducible**
**UGT2B15*2 c253G>T**	Homozygote	**Reduced activity**
**HLA-A 3101**	Absent	**No increased risk for carbamazepine-induced hypersensitivity reactions**
**HLA-B*1502**	Absent	**No increased risk of carbamazepine-induced Stevens-Johnson syndrome**
**ABCB 1 c.3435C>T**	Heterozygote	**Reduced activity**

## Data Availability

Data supporting reported results can be obtained directly in writing to the corresponding author.

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
