# Peer review of "Possible Use of Minocycline in Adjunction to Intranasal Esketamine for the Management of Difficult to Treat Depression following Extensive Pharmacogenomic Testing: Two Case Reports"

_jpm, 2022, doi:10.3390/jpm12091524_

Round 1
Reviewer 1 Report
Minocycline in adjunction to intranasal esketamine for the management of difficult to treat depression following extensive pharmacogenomic testing: two case reports
Abstract:
Well written, no changes suggested.
Introduction:
Well written, could do with a spell and grammar check.
Methods:
A brief section on how/why these cases were recruited should be included. Ethics/consent requirements should be included – this is included later, but a methods section would be useful detailing interview strategies and other methods of case assessment.
Case1:
Table 2 – a column showing the range of scores and what they mean should be included.
Table 4 – Extensive metabolizer terminology is no longer preferred – please replace this with normal metabolizer. https://www.ncbi.nlm.nih.gov/pmc/articles/PMC5253119/
· CYP3A5 – should be Intermediate metabolizer
· CYP1A2 does not indicate metabolizer status, this should be removed and left with Inducible only.
· UTG2B15 – TYPO – should be UGT2B15
· CYP2D6 – depending on which allele has an extra copy this may be a poor metabolizer.
There is no discussion of these genes with respect to previous treatment with TCAs and SSRIs – please discuss. There is some discussion later on, however a discussion here on the possibility of these genotypes associating with treatment failure should be discussed.
Case 2:
Please correct table 2 and table4 as above.
Discussion: Is good, but is very general and does not focus on the two cases.
Conclusion, similarly should be focused on the two cases. The reference to hiPSC is random and has no bearing on this manuscript.
Reviewer 2 Report
I read with interest the manuscript by Marcatili et al. where they showed two case reports of two patients who are suffering from treatment resistant depression. The two patients showed improvement upon receiving ESK and minocycline.
The manuscript is well written and also speculative in nature.
The main concern is the lack of explanation. The reason why the authors decided to use ESK in the first place is not yet clear. Secondly, it is also not clear why they decided to specifically supplement the course of treatment by adding minocycline to the course if treatment.
Also, it could be important to note if any of the two patients suffered any side effects.
Additionally, I wonder if minocycline can be administrated alone without nasal ESK to TRD patients and what would be the outcome.
The part where the authors describe the genetic variations in the patients is not clearly connected to the scope of the manuscript. Is there any difference in having a certain SNP on the outcome of the analysis.
